# Dialysis Patients Respond Adequately to Influenza Vaccination Irrespective of Dialysis Modality and Chronic Inflammation

**DOI:** 10.3390/jcm12196205

**Published:** 2023-09-26

**Authors:** Christos Pleros, Konstantinos Adamidis, Konstantia Kantartzi, Ioannis Griveas, Ismini Baltsavia, Aristides Moustakas, Antonios Kalliaropoulos, Evaggelia Fraggedaki, Christina Petra, Nikolaos Damianakis, Andreas Mentis, Eleni Drosataki, Ioannis Petrakis, Ploumis Passadakis, Periklis Panagopoulos, Kostas Stylianou, Stylianos Panagoutsos

**Affiliations:** 1Nephrology Department, University Hospital of Heraklion, 71500 Heraklion, Greece; xpleros@gmail.com (C.P.); elenidro2@hotmail.com (E.D.); petrakgia@gmail.com (I.P.); 2Dialysis Unit Bionephros, 18345 Athens, Greece; kostasadamidis@yahoo.gr; 3Nephrology Department, University Hospital of Alexandroupoli, 68100 Alexandroupoli, Greece; kokan0910@gmail.com (K.K.); ploumis@mail.duth.gr (P.P.); spanagou@med.duth.gr (S.P.); 4Nephrology Department, 417 Army Share Fund Hospital of Athens, 11521 Athens, Greece; giannisgriv@hotmail.com; 5Laboratory of Computational Biology, Division of Basic Sciences, Medical School, University of Crete, 71500 Heraklion, Greece; ibaltsavia@gmail.com; 6Infometrics Data Analytics Ltd., 128 City Road, London EC1V 2NX, UK; arismoustakas@gmail.com; 7National Influenza Reference Laboratory for Southern Greece, Hellenic Pasteur Institute, 11521 Athens, Greece; kalliaropoulos@pasteur.gr (A.K.); mentis@pasteur.gr (A.M.); 8Nephroxenia Dialysis Unit, 73100 Chania, Greece; ev.fraggedaki@gmail.com; 9Venizeleio-Pananeio General Hospital of Heraklion, 71409 Heraklion, Greece; xpetranephrologist@gmail.com; 10Nephrology Department, General Hospital of Chania, 73300 Chania, Greece; damianonikolis@hotmail.com; 11Department of Infectious Diseases, Second University Department of Internal Medicine, University Hospital of Alexandroupoli, Democritus University of Thrace, 68100 Alexandroupoli, Greece; ppanago@med.duth.gr

**Keywords:** chronic inflammation, dialysis, hemodiafiltration, immune response, influenza vaccination

## Abstract

(1) Background: Chronic inflammation and suboptimal immune responses to vaccinations are considered to be aspects of immune dysregulation in patients that are undergoing dialysis. The present study aimed to evaluate immune responses in hemodialysis (HD) and online hemodiafiltration (OL-HDF) patients to a seasonal inactivated quadrivalent influenza vaccine (IQIV). (2) Methods: We enrolled 172 chronic dialysis patients (87 on HD and 85 on OL-HDF) and 18 control subjects without chronic kidney disease in a prospective, cross-sectional cohort study. Participants were vaccinated with a seasonal IQIV, and antibody titers using the hemagglutination inhibition (HI) assay were determined before vaccination (month 0) and 1, 3, and 6 months thereafter. Demographics and inflammatory markers (CRP, IL-6, IL-1β) were recorded at month 0. The primary endpoints were the rates of seroresponse (SR), defined as a four-fold increase in the HI titer, and seroprotection (SP), defined as HI titer ≥ 1/40 throughout the study period. Statistical analyses were conducted in R (version 3.6.3) statistical software. The differences between groups were analyzed using chi-square and *t*-test analyses for dichotomous and continuous variables, respectively. To identify independent determinants of SR and SP, generalized linear models were built with response or protection per virus strain as the dependent variable and group, age, sex, time (month 0, 1, 3, 6), diabetes, IL-6, dialysis vintage, HD access, and HDF volume as independent explanatory variables. (3) Results: SR and SP rates were similar between control subjects, and dialysis patients were not affected by dialysis modality. SP rates were high (> 70%) at the beginning of the study and practically reached 100% after vaccination in all study groups. These results applied to all four virus strains that were included in the IQIV. IL-6 levels significantly differed between study groups, with HD patients displaying the highest values, but this did not affect SP rates. (4) Conclusions: Dialysis patients respond to influenza immunization adequately and similarly to the general population. Thus, annual vaccination policies should be encouraged in dialysis units. OL-HDF reduces chronic inflammation; however, this has no impact on SR rates.

## 1. Introduction

Pandemics and epidemics have afflicted mankind throughout history and have increased in frequency since the Middle Ages, when the Plague of Justinian emerged [1]. Together with antibiotics, vaccines have proven to be one of the most powerful weapons in the armamentarium of the human race in its fight against infectious diseases, as has been highlighted recently during the COVID-19 pandemic [2,3,4]. Among viral diseases in the pre-COVID-19 era, influenza dominated as a major cause of pandemics and epidemics, having a high impact on annual morbidity and mortality [5]. Seasonal influenza epidemics cause 3 to 5 million severe cases and 300.000 to 500.000 deaths globally each year according to the World Health Organization (WHO), with the highest burden of disease affecting the upper and lower extremes of age as well as affecting patients with underlying comorbidities [6].

Experimental and epidemiological data suggest a strong association between chronic kidney disease (CKD) and immune system dysfunction [7]. Chronic inflammation, altered toll-like receptor (TLR) activity, and reduced lymphoid and myeloid cell response to antigenic stimuli are key elements of immune dysregulation in end-stage renal disease (ESRD) [8,9,10,11,12]. Infectious diseases account for 20% of the total mortality in ESRD patients, being the second leading cause of death behind cardiovascular disease (CVD) [13]. Regarding influenza, ESRD patients carry a three-fold increased risk of death compared with the general population, according to a Japanese cohort study during the 2008–2009 flu season [14]. A large Fresenius cohort showed an even harder impact of influenza on the dialysis population during the 2009 H1N1 pandemic. Very high hospitalization (34%) and mortality (5%) rates were reported among infected dialysis patients, whereas analogous estimates in the general population during the same period were 6–7% and 0.4%, respectively [15].

Annual vaccination of high-risk groups against influenza, among these being patients with CKD, has been a well-established public health policy endorsed by the WHO [16]. The effectiveness of the various influenza vaccines in the general population, defined as disease prevention after vaccination, is considered to be modest, with estimates being around 40–60%, even in years when vaccines are well-matched to dominant circulating viruses [6]. Studies in the dialysis population have reported conflicting results, namely rates of seroprotection ranging from 33% to 80%, with the majority yielding a weaker immune response to the trivalent influenza vaccine compared with controls [17]. Real-world epidemiological studies towards influenza vaccine effectiveness in CKD patients are still lacking, though reduced vaccine immunogenicity in general is considered to be a characteristic feature of ESRD based on findings regarding hepatitis B, tetanus, diphtheria, pneumococcal, and COVID-19 vaccination [18,19,20,21,22,23,24].

Besides the uremic milieu itself, different dialysis modalities appear to have a significant impact on the chronic inflammation and immune dysfunction of ESRD patients. Hemodialysis (HD) has been associated with elevated endotoxin serum levels and decreased neutrophil phagocytic activity compared with peritoneal dialysis (PD) [25]. Online hemodiafiltration (OL-HDF) attenuates HD-associated systemic inflammation, reducing proinflammatory CD14+ CD16+ monocyte-derived dendritic cells and cytokine levels in the serum of dialysis patients [26,27]. Several observational studies and four randomized controlled trials (RCT) suggest an overall survival benefit from the implementation of high-volume OL-HDF as opposed to conventional low- or high-flux HD [28,29]. A French RCT demonstrated improved treatment tolerance of OL-HDF but no effect on morbidity and mortality compared with high-flux HD in elderly patients [30].

The present study aimed to investigate the overall response of dialysis patients to vaccination with an inactivated quadrivalent influenza vaccine (IQIV) and explored the possibility of a better immunological profile of OL-HDF patients.

## 2. Materials and Methods

### 2.1. Study Design and Participants

A prospective, cross-sectional cohort study was conducted in chronic dialysis patients from 5 dialysis units in Greece. The primary objective of this study was to assess patients’ immune response to annual influenza vaccination and address possible differences related to different dialysis modalities and chronic inflammation. We enrolled 172 chronic dialysis patients (87 on HD and 85 on OL-HDF treatments) from 5 dialysis units across Greece and 18 controls without CKD, mainly hospital personnel and patients’ relatives; all of them were aged 18 years or older, and they were recruited before the 2016–2017 winter season. Patients receiving dialysis therapy for less than 2 months and participants with a diagnosis of cancer during the previous 5 years were excluded. Furthermore, we excluded participants with symptoms that were suggestive of active infection, a history of allergy to influenza vaccine or eggs, and those who were taking corticosteroids or other immunosuppressive medication in the last 12 months prior to the beginning of the study.

Annual influenza vaccination is part of the standard policy that is followed by all of the participating dialysis units. All individuals provided written informed consent to participate in the study.

### 2.2. Influenza Vaccination

All participants received a single standard dose of IQIV (Vaxigrip Tetra/Sanofi, Pasteur, Europe) intramuscularly. According to the recommendations of the WHO and European Medicines Agency (EMA), the 2016 vaccine contained 15 mcg per strain of A/California/7/2009 (H1N1)pdm09-like virus, A/Hong Kong/4801/2014 (H3N2)-like virus, B/Brisbane/60/2008-like virus (Victoria lineage), and B/Phuket/3073/2013-like virus (Yamagata lineage) [31,32].

### 2.3. Demographics and Clinical Data

The following information was extracted from the participants’ medical files: age, sex, diabetic status, dialysis modality, dialysis vintage, HD access, and HDF substitution volume. Blood samples were drawn prior to influenza vaccination (month 0) and 1, 3, and 6 months post-vaccination. Samples were centrifuged, and sera were stored at −20 °C until analysis. The serum levels of C-reactive protein (CRP) were measured in all participants at month 0 and were thereafter monitored throughout the study period following the standard practice of each dialysis unit. CRP measurements outside of the predefined timepoints (month 0, 1, 3, 6) were excluded from the final data analysis. The serum levels of IL-1β and IL-6 were determined in duplicate in all participants at month 0 using ELISA (R&D systems Inc., Bio-Techne Ltd.—Quantikine ELISA Human IL-1β/IL-1F2 & IL-6).

### 2.4. Influenza Antibody Titers and Outcome Measures

Antibody titers at month 0, 1, 3, and 6 were determined in duplicate using the hemagglutination inhibition (HI) assay, and the values were expressed as dilutions ranging from 1:10 up to 1:1280. Serological assays were performed at the Department of Microbiology of the Hellenic Pasteur Institute, which is the national reference laboratory for Southern Greece.

The primary endpoints were the rates of seroresponse (SR) and seroprotection (SP) that were observed throughout the 6-month study period according to international definitions: (1) SR was defined as a post-vaccination HI titer ≥ 1:40 with a pre-vaccination HI titer ≤ 1:10, or a minimum four-fold increase in post-vaccination HI antibody titer. Only HI antibody titers at month 1 post-vaccination were used for SR characterization. (2) SP was defined as an HI antibody titer ≥ 1:40 [33,34], which is considered the 50% protective threshold, beyond which it is unlikely that serious clinical illness will occur in immunocompetent persons [35]. The 6-month timeframe was chosen so as to assess HI antibody kinetics across the whole period when influenza outbreaks typically occur. All plots were generated in the R environment (R version 3.6.3) using the ggplot2 (version 3.4.2) and cowplot (version 1.1.1) packages. Additional re-grouping and filtering of initial data for the use of them in plots was conducted using the dplyr (version 1.1.2) package. The seroresponse rates were visualized in bar plots as the percentage of responders in each group of patients for each strain. An alluvial plot of seroprotection and seroresponse was generated by counting the number of protected and non-protected patients at time-point month 0, as well as which of them became responders or non-responders at time-point month 1 for each strain among all groups of patients. Antibody titers were first log-transformed (with base 10) and then plotted, calculating the mean and standard deviation (s) of the sample (denominator: n-1) at each time point (months 0, 1, 3, 6) using the functions mean and sd, respectively, from the stats (version 3.6.3) package.

### 2.5. Statistical Analysis

All analyses were conducted in R statistical software (version 3.6.3) [36]. HI titers were converted to log_10_ values to express the kinetics throughout the 6-month study period. Differences between groups were analyzed using chi-square and *t*-test analyses for dichotomous and continuous variables, respectively. A significance level of a two-sided *p* = 0.05 was set. To identify independent determinants of SR and SP, generalized linear models (GLMs) were built with response or protection per virus strain as the dependent variable (H1N1, H3N2, Yamagata, and Victoria; each were replicated by response and protection efficacy) and group (Control—HD—HDF), age, sex, time (month 0, 1, 3, 6), diabetes, IL-6, dialysis vintage (months), HD access, and HDF volume as independent explanatory variables. The model was fitted with a binomial family error structure. In all cases, model residuals were inspected for heteroscedasticity, and we inspected that all assumptions of ANOVA were successfully met.

## 3. Results

### 3.1. Study Population

The demographic and clinical characteristics of all participants and subgroups are listed in Table 1. HD patients were substantially older (*p* < 0.01) than the rest of the study population, having a higher incidence of diabetes (*p* < 0.05) and male sex (*p* < 0.05). HDF patients were on dialysis for a longer period (*p* < 0.001) and were dialyzed through an AV fistula more frequently than HD patients (*p* < 0.05). CRP levels did not significantly differ among participants, whereas IL-6 levels were significantly lower in control subjects (*p* < 0.001). Among dialysis patients, HDF was associated with lower IL-6 levels compared with HD patients (*p* = 0.011). With the exception of two participants, IL-1β was consistently undetectable in serum samples; hence, it was omitted from the final data analysis. One participant in the HDF group died before the serum sampling at month 1 from a cause unrelated to the IQIV and was therefore excluded from the study.

### 3.2. Immune Response to Influenza Vaccination

The SR and SP rates are listed in Table 2. SR rates 1 month after vaccination did not significantly differ between groups for any of the four virus strains of the seasonal IQIV (*p* > 0.05) (Figure 1). Higher response rates were observed for Yamagata and Victoria B strains (66.3%) compared with H1N1 and H3N2 A strains (57.5% and 31.7%, respectively). Baseline protection rates were > 90% for all virus strains except for H1N1 (protection rate 81.3%) and were comparable between groups.

After vaccination, only one participant in the HDF group remained unprotected for the H3N2 strain. Throughout the 6-month study period, all other subjects generated protective antibody titers against all strains of the seasonal IQIV.

Participants who were originally not protected (with a pre-vaccination HI titer < 1/40), displayed 100% SR and SP rates post-vaccination and throughout the study period, regardless of study group (Figure 2). HI antibody titers, which were expressed as log_10_ values, were similar between groups throughout the whole study period (Figure 3). The time of HI titer measurement (month 0, 1, 3, 6) was the only parameter that was strongly associated with the protection rate for all viral strains in the multivariate analysis (*p* < 0.001).

Older age was negatively associated with protection solely for the H3N2 strain (*p* = 0.039). The presence of diabetes was associated with a higher SR to H1N1 and Victoria strains (*p* < 0.001). Dialysis vintage was positively correlated with H1N1 SR (*p* = 0.014) and negatively correlated with Victoria SR (*p* = 0.004). SR to H3N2 and Yamagata strains was significantly associated with HDF (*p* = 0.006 and *p* = 0.001 respectively) and higher IL-6 levels (*p* < 0.001).

With respect to the H3N2 strain in particular, female sex (*p* = 0.001) and the presence of an AV fistula (*p* = 0.008) conferred a higher SR rate, whereas a higher HDF substitution volume was negatively associated with SR (*p* < 0.001) (Appendix A).

## 4. Discussion

Dialysis patients’ immune response to influenza vaccination has been a matter of controversy over the past decades. Our study aimed to explore the immunogenicity of an IQIV vaccination in dialysis patients and highlight the potential differences related to HD modality. Between control subjects and dialysis patients, as well as between HD and HDF, there was no difference in SR and SP rates. Although HD patients’ cytokine levels were significantly higher than those of the other participants, as was to be expected, this did not translate into a poorer immunological response to influenza vaccination. Contrary to our expectations, higher IL-6 levels were linked to a better SR to the H3N2 and Yamagata strains. SP rates were very high prior to immunization, and vaccine delivery led to an almost 100% protection of the study population throughout the 6-month trial period. SR to B strains (Yamagata and Victoria) of the IQIV appeared superior compared with SR to A strains (H1N1 and H3N2), with H3N2 displaying the lowest SR rate.

ESRD and dialysis patients are considered immunocompromised, and infections represent a major cause of morbidity and mortality in this population [13]. Poor immune responses to immunization against hepatitis B, diphtheria, tetanus, pneumococcus, and COVID-19 have been reported, and the results were interpreted within the aforementioned context. In general, dialysis patients are thought to be incapable of eliciting a sufficient immune response to antigenic stimuli [19,23,24,37,38,39,40]. Traditionally, an adequate immune response to influenza vaccination has been defined as a four-fold increase of antibodies towards hemagglutinin, an antigen expressed on the surface of influenza viruses. This definition has been incorporated in the methodology of the most important studies in the field. Studies about influenza vaccination in the dialysis population have used various vaccine formulations and schemes, and the results on SR and SP rates have differed considerably on numerous occasions. Our study did not provide evidence to support the hypothesis that dialysis patients would not respond adequately to influenza immunization. Surprisingly, SR was comparable between the three groups of participants and comparable to the current literature on SR rates in healthy adults under the age of 65 years [41]. Similar to this, Sharpe et al. observed no difference in SR between HD patients and healthy volunteers in one of the largest studies in this field, although they observed very low SR rates overall [42]. It must be stressed that large variations in SR among dialysis patients documented so far may be the result of selection bias and diverse methodology, as adequately powered prospective randomized control trials have not been carried out to date. An interesting finding was the 100% SR rate among participants who were not protected (HI titer < 1/40) at the beginning of the study. We speculate that some of them might have been unvaccinated against influenza in the previous season and therefore display a robust immune response after immunization, as has been documented by Keitel et al. [43].

Chronic inflammation, as reflected in serum IL-6 levels, significantly differed between the groups of participants in our study but did not correlate with an inferior SR or SP rate. On the contrary, higher IL-6 levels were linked to a better SR against the H3N2 and Yamagata strains. Chronic inflammation is a well-known feature of CKD and dialysis patients and has been linked to the progression of atherosclerosis and subsequent cardiovascular morbidity and mortality [13,44]. Our original hypothesis that HDF would be linked to reduced inflammation and better SR at the same time, which would provide a plausible relationship between chronic inflammation and poor immunological responses, proved to be incorrect. In our opinion, the fact that participants with higher IL-6 levels responded better against two virus strains of the seasonal IQIV could be attributed to the insufficient power of our study sample. The same hypothesis may also explain the contradictory or inconclusive results regarding the effect of other parameters (age, diabetes, dialysis vintage, sex, AV fistula) on SR to specific virus strains.

The immune response to influenza, like most viral diseases, involves both cellular and humoral mechanisms. HI titers, though not directly correlated with protection against influenza illness, are considered to be a surrogate of protection [45]. The likelihood of infection progressively declines with increasing HI titer [46] results, and a cutoff of 1/40 is proposed by regulators as being protective, a compromising correlation for immunogenicity trials conducted by influenza vaccine manufacturers for licensing purposes [34,47]. Very high pre-vaccination SP rates were observed in our study cohort, and vaccination resulted in a universal SP against all virus strains of the seasonal IQIV. Although this is encouraging, our results probably highlight the weak correlation between HI titer and actual protection against clinical infection. To address this issue, regulators may need to revise their policies regarding the approval of seasonal influenza vaccines. For this reason, Manley et al. chose to consider HI titers ≥ 1/160 as protective while conducting an immunogenicity study in dialysis patients, which resulted in a lower SP rate [48]. Furthermore, past immunizations are known to affect baseline HI titers [46], and this may have accounted for the high SP rate observed in our study cohort.

There are some limitations in this study that need consideration. First, this was a non-randomized study that lacked sufficient power to show a significant difference in the immune response between patients on dialysis and participants without CKD. Second, the number of control subjects was very low compared with the other two groups. This was an intentional choice, considering our limited financial budget, so as to preferentially investigate the immune responses of HD and OL-HDF patients. Nevertheless, our results did not differ between groups and were in concordance with the current literature on influenza vaccination in the general population. Third, because no documented influenza illness tracking was performed, conclusions about protection status should be regarded as simple hypotheses. Fourth, past influenza vaccination history was not recorded, and this may have led to an underestimation of SR [43]. Based on the dialysis units’ policy, yearly influenza vaccination was assumed as a standard of care for the dialysis patients enrolled in the study. Aside from the limitations noted above, this study has two significant strengths. It is one of the largest studies about influenza vaccination in dialysis patients and, to the best of our knowledge, it is the largest study with a direct comparison of immune responses between HD and OL-HDF patients. The clinical implication is that dialysis patients can be effectively immunized against influenza regardless of hemodialysis modality or chronic inflammation status.

Conclusions: This study found that dialysis patients have high SR and SP rates, which supports annual vaccination policies. Despite being an important determinant of chronic inflammation, HD modality does not appear to alter patients’ immune responses to antigenic stimuli. Dialysis patients respond to influenza vaccination in the same way that the general population does.

## Figures and Tables

**Figure 1 jcm-12-06205-f001:**
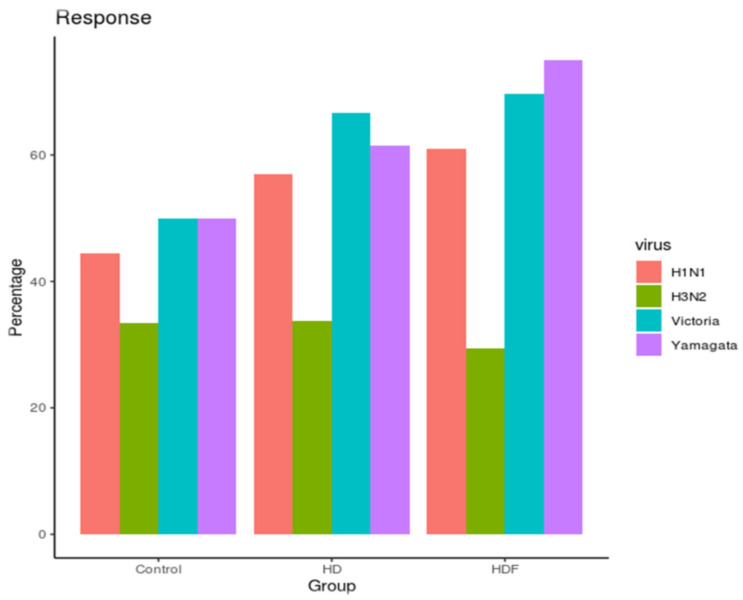
Seroresponse rates among groups of participants. Abbreviations: HD, hemodialysis; HDF, hemodiafiltration.

**Figure 2 jcm-12-06205-f002:**
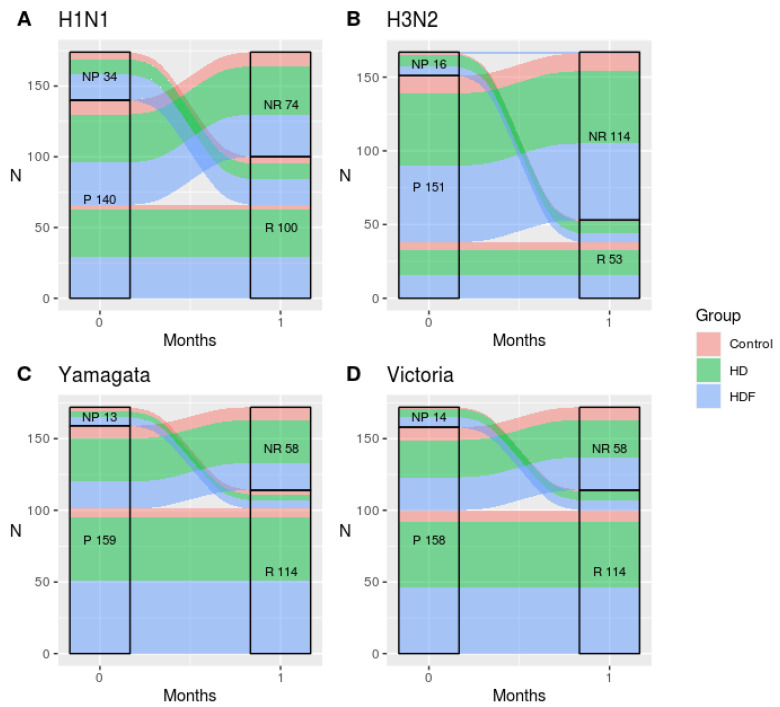
Alluvial plot of seroprotection and seroresponse correlation among groups of participants. Abbreviations: NP, non-protected; P, protected; NR, non-responders; R, responders; HD, hemodialysis; HDF, hemodiafiltration; N, number of participants. The definitions of seroresponse and seroprotection can be seen in Section 2.

**Figure 3 jcm-12-06205-f003:**
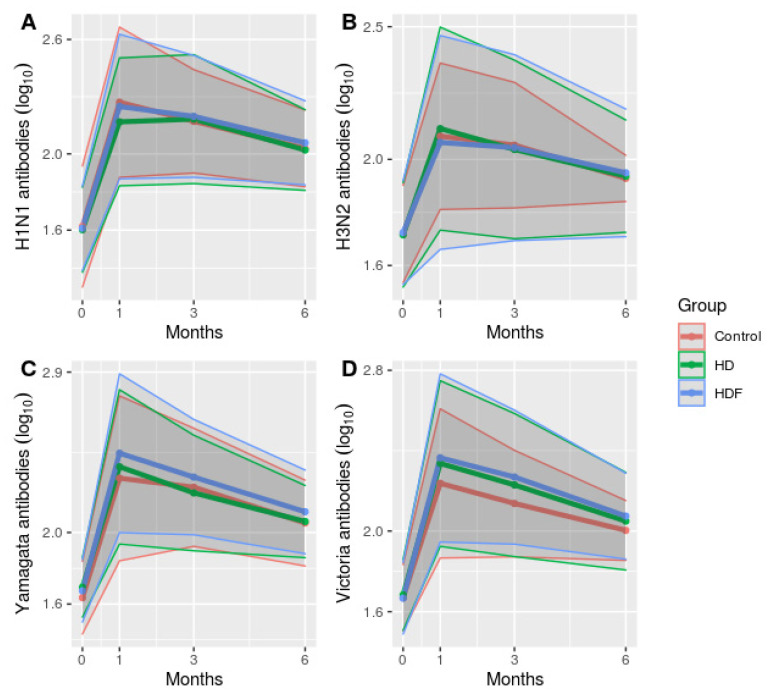
Plot of antibody titers evolution after vaccination (expressed as log_10_ values) ± SD throughout study period. Abbreviations: HD, hemodialysis; HDF, hemodiafiltration. Antibody titers were first log-transformed (with base 10) and then plotted by calculating the mean and standard deviation of the sample (denominator: n-1) at each time point (month 0, 1, 3, 6). (**A**) H1N1 antibody titers, (**B**) H3N2 antibody titers, (**C**) Yamagata antibody titers, (**D**) Victoria antibody titers.

**Table 1 jcm-12-06205-t001:** Participant characteristics.

	Total	Control	HD	HDF	*p*-Value HD vs. Control	*p*-Value HDF vs. Control	*p*-Value HD vs. HDF
*N*	189	18	87	84			
Age	62.6 ± 13.4	60.9 ± 11.6	65.3 ± 13.4	60.2 ± 13.3	<0.05	NS	<0.05
Male/Female	122/67	6/12	63/24	53/32	<0.05	<0.05	<0.05
Diabetes	50 (26.5%)	2 (12.5%)	30 (34.5%)	19 (22.6%)	<0.05	<0.05	<0.05
Dialysis vintage (months)		NA	45.1 ± 41.3	62 ± 61.4	NA	NA	<0.05
CVC		NA	19 (21.8%)	6 (7.1%)	NA	NA	<0.05
Fistula		NA	49 (56.3%)	63 (75%)	NA	NA	<0.05
Graft		NA	18 (20.7%)	15 (17.9%)	NA	NA	NS
HDF volume (L)		NA	NA	19.79 ± 4.06	NA	NA	NA
CRP	0.969 ± 1.58	0.331 ± 0.337	0.995 ± 1.567	0.987 ± 1.649	NS	NS	NS
IL-6	5.81 ± 10.15	0.76 ± 2.2	7.53 ± 10.76	5.27 ± 10.25	<0.05	<0.05	<0.05

Note: Values are presented as mean ± SD. Abbreviations: HD, hemodialysis; HDF, hemodiafiltration; CVC, central venous catheter; CRP, C-reactive protein; IL-6, interleukin 6; NA, not applicable; NS, not significant.

**Table 2 jcm-12-06205-t002:** Seroresponse and seroprotection rates.

	Control (%)	HD (%)	HDF (%)	Total (%)	P
**H1N1**					
SR	44.4%	57%	61%	57.5%	0.43
SP-month 0	72.2%	87.1%	77.2%	81.3%	0.16
SP-month 1	100%	100%	100%	100%	NA
SP-month 3	100%	100%	100%	100%	NA
SP-month 6	100%	100%	100%	100%	NA
**H3N2**					
SR	33.3%	33.8%	29.3%	31.7%	0.83
SP-month 0	94.4%	90.6%	91.1%	91.2%	0.87
SP-month 1	100%	100%	98.7%	99.4%	0.54
SP-month 3	100%	100%	98.7%	99.4%	0.52
SP-month 6	100%	100%	98.6%	99.4%	0.55
**Yamagata**					
SR	50%	61.5%	75.0%	66.3%	0.06
SP-month 0	83.3%	95.3%	92.4%	92.9%	0.2
SP-month 1	100%	100%	100%	100%	NA
SP-month 3	100%	100%	100%	100%	NA
SP-month 6	100%	100%	100%	100%	NA
**Victoria**					
SR	50%	66.7%	69.7%	66.3%	0.28
SP-month 0	94.4%	92.9%	91.1%	92.3%	0.85
SP-month 1	100%	100%	100%	100%	NA
SP-month 3	100%	100%	100%	100%	NA
SP-month 6	100%	100%	100%	100%	NA

Note: Values expressed as percentage of participants. Abbreviations: HD, hemodialysis; HDF, hemodiafiltration; SR, seroresponse; SP, seroprotection; NA, not applicable.

## Data Availability

The data presented in this study are available in Appendix A here.

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
