# Peer review of "Dialysis Patients Respond Adequately to Influenza Vaccination Irrespective of Dialysis Modality and Chronic Inflammation"

_jcm, 2023, doi:10.3390/jcm12196205_

Round 1

Reviewer 1 Report

Dialysis patients respond adequately to influenza vaccination, irrespective of dialysis modality and chronic inflammation.

Thank you very much for allowing me to review this manuscript. Here are some comments to improve it.

Abstract:

• More information about the methodology should be indicated in the abstract. The current one is insufficient.

Methodology:

• I see an imbalance between the number of people participating in the control group and the intervention group. This makes it difficult to compare the results in both cases and can lead to a bias in the results.

• Indicate what the exclusion criteria are.

Results:

• Indicate the p-values in Table 1.

• Provide the confidence intervals in Table 2.

Discussion:

• Include the strengths of the study and the implications for clinical practice.

Conclusions:

• Incorporate the conclusions section.

Author Response

Reviewer 1

Thank you very much for allowing me to review this manuscript. Here are some comments to improve it.

1) Abstract:

  • More information about the methodology should be indicated in the abstract. The current one is insufficient.

Response 1.  Methodology in the abstract has been enriched by including statistical methods used, as suggested by the reviewer

2) Methodology:

2A) • I see an imbalance between the number of people participating in the control group and the intervention group. This makes it difficult to compare the results in both cases and can lead to a bias in the results.

Response 2A. Indeed, the number of control subjects was low compared to the other two groups. This is already covered in the "limitations" section. Given our limited financial resources, we made the decision to focus on the immune responses of HD and OL-HDF patients. Nonetheless, our results in the control group were consistent with previously published data on influenza vaccination in the general population. As a result, we do not anticipate any deviation from the current results if more subjects were included in the control group

2B) • Indicate what the exclusion criteria are.

Response 2B. Exclusion criteria were described in the last paragraph in page 2 as follows: “Patients receiving dialysis therapy for less than 2 months and participants with a diagnosis of cancer during the previous 5 years were excluded. Furthermore, we excluded participants with symptoms suggestive of active infection, a history of allergy to influenza vaccine or egg, and those who were taking corticosteroids or other immunosuppressive medication in the last 12 months prior to the beginning of the study”.

3) Results:

3A) • Indicate the p-values in Table 1.

Response 3A:  The p values are shown at the footnote of the Table 1 as follows: * p<0.05 between HD and Control; § p<0.05 between HDF and Control; † p<0.05 between HD and HDF.

3B) • Provide the confidence intervals in Table 2.

Response 3B. Values in table 2 are binary. Therefore, there are no confidence intervals.

4) Discussion:

  • Include the strengths of the study and the implications for clinical practice.

Response 4: the following paragraph was added: “this study has two significant strengths. It is one of the largest studies about influenza vaccination in dialysis patients, and to the best of our knowledge, the largest study with a direct comparison of immune responses between HD and OL-HDF. The clinical implication is that dialysis patients can be effectively immunized against influenza without regard to hemodialysis modality or chronic inflammation status”.  

5) Conclusions:

  • Incorporate the conclusions section.

Response 5: conclusions were included at the end of the manuscript as follows: “Conclusions: This study found that dialysis patients have high SR and SP rates, which supports annual vaccination policies. Despite being an important determinant of chronic inflammation, HD modality does not appear to alter patients' immune responses to antigenic stimuli. Dialysis patients respond to influenza vaccination in the same way that the general population does”.

On behalf of my co-authors

Kind regards

The corresponding author

Reviewer 2 Report

Please see comments below:

Line 209-210, as authors stated “higher IL-6 levels were linked to a better SR to H3N2 and Yamagata strains”, no data was shown in the main context. As indicated in the method part (Line115), IL-6 and IL-1beta were only measured at month 0, would authors consider including measurements on more timepoints after vaccination to better support the statement mentioned above?

Groups are differed significantly in age, sex or diabetes status, of which the potential impact on the conclusion should be discussed.

High baseline protection (SP rate) was detected pre-vaccination, and authors questioned the connection between HI titer and real protection in the discussion, would authors consider modifying/optimizing threshold that defines protection which is also mentioned in line 263 by Manley group?

Missing definition on responder and non-responder for figure 2

Please specify analysis method for figure 3

Results described from line 191-201 were not clearly linked with corresponding data sets.

Supp data is poorly organized with no clear figure labeling or legend information.

Author Response

Reviewer 2

Comment 1A) Line 209-210, as authors stated “higher IL-6 levels were linked to a better SR to H3N2 and Yamagata strains”, no data was shown in the main context.

Comment 1B) As indicated in the method part (Line115), IL-6 and IL-1beta were only measured at month 0, would authors consider including measurements on more time points after vaccination to better support the statement mentioned above?

Response 1A:  These data were given in a supplementary file. We have now improved the clarity and readability of the supplementary file.  We appreciate the reviewer's feedback.

Response 1B: The timing of IL1 and IL6 measurements was chosen to be as close to the vaccination date as possible. More distant measurements would be irrelevant and unlikely to influence vaccination response. They would also raise the study's cost. As a result, we did not measure ILs levels at other time points. We do not believe that remote measurements would have influenced the results because antibody titres increased sharply during the first month (response), and then declined uniformly and slowly over the next 6 months (see figure 3).

Comment 2) Groups differ significantly in age, sex or diabetes status, of which the potential impact on the conclusion should be discussed.

Response 2:

In most cases, the multivariate analysis revealed no significant differences between groups (demographics, modalities, comorbidities). The few statistically significant differences discovered were all mentioned in the results section. These, however, were inconsistent across strains and occasionally contradictory. As a result, we stated the following in the discussion section: “In our opinion, the fact that participants with higher IL-6 levels responded better against two virus strains of the seasonal IQIV could be attributed to the insufficient power of our study sample. The same hypothesis may also explain the contradictory or inconclusive results regarding the effect of other parameters (age, diabetes, dialysis vintage, sex, AV fistula) on SR to specific virus strains”.

There might also exist another explanation for the inconsistencies observed, that was mentioned in the limitations section. Past influenza vaccination history was not recorded and was taken for granted for HD and OL-HDF groups due to dialysis units’ uniform policies. However, this may not be absolutely true, and may have led to underestimation (in cases of recent vaccination) or overestimation (in cases of remote vaccination) of SR among groups.

Comment 3) High baseline protection (SP rate) was detected pre-vaccination, and authors questioned the connection between HI titer and real protection in the discussion, would authors consider modifying/optimizing threshold that defines protection which is also mentioned in line 263 by Manley group?

Response 3: This is an intriguing point brought up by the reviewer. However, in order to be consistent with the vast majority of other studies in the field, we chose to use WHO thresholds that have been in place for decades.

Comment 4) Missing definition on responder and non-responder for figure 2.

Response 4: these definitions were given in the methods section as follows: “SR was defined as a post-vaccination HI titer ≥1:40 with a pre-vaccination HI titer ≤1:10, or a minimum four-fold increase in post-vaccination HI antibody titer. Only HI antibody titers at month 1 post-vaccination were used for SR characterization. 2) SP was defined as an HI antibody titer ≥1:40, which is considered the 50% protective threshold, beyond which it is unlikely that serious clinical illness will occur in immunocompetent persons”.

Thus, we simply added a comment in Figure 2 footnote stating: the definitions of seroresponse and seroprotection can be seen in the “MATERIALS AND METHODS” section

Comment 5) Please specify analysis method for figure 3

Response 5: We added in the M&M section the following paragraph so as to specify the analysis method for figures: “All plots were generated in the R environment (R version 3.6.3) with packages ggplot2 (version 3.4.2) and cowplot (version 1.1.1). Additional re-grouping and filtering of initial data so as to use them in plots, was conducted with package dplyr (version 1.1.2). Seroresponse rates were visualized in barplots, as the percentage of responders in each group of patients for each strain. Alluvial plot of Seroprotection and Seroresponse was generated by counting the number of protected and Non-Protected patients in time-point month 0 and which of them became Responders or Non-Responders at time-point month 1 for each strain, among all groups of patients. Antibody titers were first log transformed (with base 10) and then plotted calculating the mean and standard deviation (s) of the sample (denominator: n-1) at each time point (months 0, 1, 3, 6) using functions mean and sd respectively from the package stats (version 3.6.3)”.

Comment 6) Results described from line 191-201 were not clearly linked with corresponding data sets. Supp data is poorly organized with no clear figure labelling or legend information.

Response 6:  We appreciate the reviewer's feedback. The supplementary file was undoubtedly large and haphazard. We have mistakenly uploaded the whole results output (79 pages) instead of correct supplement (12 pages).   The clarity and readability of the supplementary file has now been improved. Significant differences are flagged with asterisks.

On behalf of my co-authors

Kind regards

The corresponding author

Round 2

Reviewer 1 Report

Thanks to the authors for their work. The article has improved and the authors have resolved my comments. However, there is one issue that has not been resolved: Please enter in Table 2 the p values of the relationships between the variables and the established groups. Include p-values

Author Response

WE THANK THE REVIEWER FOR THE CONSTRUCTIVE COMMENTS

p values have now been incorporated in Table 1 (Table 2 already had the p-values incorporated)

Reviewer 2 Report

Comments have been properly addressed. 

Author Response

We very much thank the reviewer for his time , effort and constructive comments